# Enhancing Bean (*Phaseolus vulgaris* L.) Resilience: Unveiling the Role of Halopriming against Saltwater Stress

Ilaria Borromeo [1], Fabio Domenici [2], Cristiano Giordani [3,4], Maddalena Del Gallo [5] and Cinzia Forni [6,*]

1 PhD School in Evolutionary Biology and Ecology, Department of Biology, University of Rome Tor Vergata, 00133 Rome, Italy; ilaria18scv@hotmail.it
2 Department of Chemical Science and Technologies, University of Rome Tor Vergata, Via della Ricerca Scientifica, 00133 Rome, Italy; fabio.domenici@uniroma2.it
3 Instituto de Física, Universidad de Antioquia, Calle 70 No. 52-21, Medellín 050010, Colombia; cristiano.giordani@udea.edu.co
4 Grupo Productos Naturales Marinos, Facultad de Ciencias Farmacéuticas y Alimentarias, Universidad de Antioquia, Calle 70 No. 52-21, Medellín 050010, Colombia
5 Department of Health, Life and Environmental Sciences, University of L'Aquila, Via Vetoio, Coppito 1, 67100 L'Aquila, Italy; maddalena.delgallo@univaq.it
6 Department of Biology, University of Rome Tor Vergata, Via della Ricerca Scientifica, 00133 Rome, Italy
* Correspondence: forni@uniroma2.it

**Abstract:** Common bean (*Phaseolus vulgaris* L.), one of the most important cultivated legumes, requires a high level of water. It is included among the most sensitive species to climate change; drought and salinity cause a reduction in photosynthesis, metabolic and enzymatic alterations, and oxidative stress. To improve crop tolerance to salt, seed priming and acclimation can be useful tools. To test the salt tolerance of beans, a preliminary screening was undertaken on four cultivars of *P. vulgaris* (Black Turtle, Cargamanto, Bola Roja, Borlotto) by exposing the seeds to different levels of salinity. The salt-sensitive cultivar Borlotto was chosen for experimental greenhouse trials to study the effects of halopriming and acclimation. Primed and non-primed seeds were sown in non-saline soil and acclimated for 2 weeks; then, the plants were watered with non-saline and saline solutions for 4 weeks. At the end of this growth period, the primed plants showed a marked increase in salt stress tolerance, improving the chlorophyll content, phenolic compounds, and many enzymes' activities, in turn reducing the effect of salt on growth and fruit production compared to the non-primed controls. In conclusion, halopriming can be considered a useful tool to enhance salinity tolerance in beans and other salt-sensitive crops.

**Keywords:** salt acclimation; halotolerance; *Phaseolus vulgaris*; saltwater stress; seed priming

## 1. Introduction

The common bean (*Phaseolus vulgaris* L.) is considered among the most important legumes for human nutrition, especially in Latin America and Eastern and Southern Africa [1]. Beans are an economical and widely available crop, and they are a source of proteins, slow-digesting carbohydrates, vitamins, minerals (i.e., iron and zinc), fiber, starch, and phytochemicals with a multitude of bioactive properties [2–4]. In developed countries, common beans are also important eco-friendly sources of protein compared to animal protein [2,3,5]; they have a unique nutritional profile due to their high protein content, which ranges between 17 and 30% of dry weight [2,3].

The inclusion of beans in the diet reduces the incidence of cancers of the gastro-intestinal tract, cardiovascular diseases, and type-2 diabetes, increases the diversity of the gut microbiota, and promotes colon health by reducing inflammatory states [2,3,6,7].

Besides health benefits, beans are also economically useful, especially in developing countries, where women are greatly involved in their cultivation, processing, marketing,

and cooking [3,8]. Being widely grown under various climatic conditions, common beans are exposed to climate change and biotic and abiotic stresses that decrease their productivity to 20% or less of the potential yield [9]. Abiotic stresses include different stresses, such as heat and cold, as well as drought and salinity stress, which are particularly harmful to plant growth [10,11]. This species is a glycophyte that is very sensitive to soil salinity, which is one of the greatest threats to bean production since it negatively affects germination, crop vigor, and yield [12]. According to Machado and Serralheiro [13], the threshold of salt tolerance in bean is 1 dS/m for soil salinity and 0.7 dS/m for irrigation water.

In saline conditions, the uptake of $Na^+$ and $Cl^-$ ions induces cytotoxicity; in addition, salinity induces oxidative stress caused by the production of reactive oxygen species (ROS) [14–16]. Other consequences of excessive salinity are water stress, reduced photosynthesis, metabolic changes, osmotic and ionic stress, decreased cell division and expansion, membrane peroxidation, DNA damage, and the inactivation of many antioxidant enzymes [17,18]. The decline in growth is also due to the nutritional imbalance caused by the competition of $Na^+$ and $Cl^-$ with other elements (i.e., nitrate, sulphate, phosphate), which reduces uptake and transport to the leaves [19]. After prolonged exposure to salt, plants become more susceptible to early senescence, which results in the premature death of the entire plant, leading to crop loss and economic damage [20,21].

Several techniques are used to increase salt tolerance in crops, including acclimation and seed priming. Stress acclimation is achieved by gradually exposing plants to stress conditions, leading to a better adaptation of plants to stress [22,23]. It is defined as the set of phenotypic changes based on molecular and physiological adjustments developed by the plant against a stressor. Plant response requires long activation times and decays if the stress is removed [24,25]. The physiological basis of this process is still unclear; thus, acclimation is based on complex and dissimilar mechanisms, these being different not only among plant species but sometimes also within the same species. Nevertheless, researchers agree about the beneficial effects of this phenomenon on plants, which include increased growth and tolerance and a reduction in leaf chlorosis [24–26]. Besides acclimation, seed priming has been established as an important method to improve plant response to various stresses. Seed priming is reported as a pre-sowing treatment, where the seeds are soaked in a priming solution, followed by their drying to avoid radicle emergence [27]. Primed seeds exhibit faster and synchronized germination and better seedling development than non-primed seedlings [27]. Priming treatment can induce abiotic stress in seeds while also providing a cross-tolerance to different abiotic stresses [28,29]. This approach was applied successfully to develop stress tolerance in several crops [27–31], even though different priming protocols have been applied, depending on the species.

This work aimed to assess the possibility of applying the seed priming approach to beans to improve their tolerance to saline conditions. Based on their distribution, importance in the traditional cuisine of Central and South America, and economic value, four cultivars of *P. vulgaris* were tested: Black Turtle, Bola Roja, Cargamanto, and Borlotto, the latter being widely used in Italy. Among these varieties, the cultivar Borlotto was chosen. Preliminary experiments were performed to determine the best priming agent and protocol, as well as the threshold of salt tolerance, and to identify the response to saline soil of primed plants. Based on previous studies [29,32] and the results obtained in these experiments, priming per se was not sufficient to overcome salinity stress when the seeds were sown in saline soils. Therefore, we decided to apply to the beans a double treatment (i.e., seed halopriming followed by acclimation to salt). The evaluation of the efficacy of halopriming and acclimation was performed by detecting the morphological and physiological responses, fruit production, and nutritional quality of the seeds.

## 2. Materials and Methods

All reagents were analytical grade or equivalent and bought from Merck KGaA, Darmstadt, Germany. During all experiments, the working solutions were freshly prepared before use.

### 2.1. Plant Material and Determination of Seeds' Halotolerance

Seeds of *Phaseolus vulgaris* L., cv. Black Turtle, Bola Roja, and Cargamanto, were kindly supplied by Prof. Cristiano Giordani of the Universidad de Antioquia of Medellín, Colombia. Seeds of *P. vulgaris* L., cv. Borlotto, were bought in a local store in Rome.

The halotolerance of the seeds (Figure 1) before and after priming treatment, was evaluated by a dose–response curve. Ten seeds were put in Petri dishes, on filter paper, embedded with 15 mL of water or with salt solutions at different concentrations (0 mM NaCl, 40 mM NaCl, 80 mM NaCl, and 160 mM NaCl). Germination rates were recorded after 7 days. The concentration of NaCl that significantly reduced seed germination was considered as threshold of salinity tolerance.

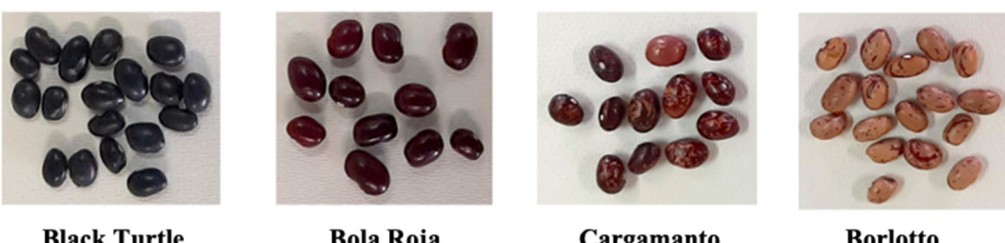

**Black Turtle**  **Bola Roja**  **Cargamanto**  **Borlotto**

**Figure 1.** Seeds of the 4 cultivars of *P. vulgaris* L.

### 2.2. Plants' Growth Conditions and Saline Treatments

The seeds were kept in the dark at room temperature (RT) until the priming treatment. Before priming, the seeds were surface-sterilized (with 70% ethanol for 5 min and then soaked in a solution of 1% NaClO for 5 min) and rinsed in double-distilled water.

To determine the best concentration of priming solution and treatment time to avoid radicle emergence, preliminary experiments were performed. Based on these tests, 40 mM NaCl was detected as the best priming agent. Seeds were plunged in 100 mL of 40 mM NaCl for 24 h at RT. At the end of the treatments, they were rinsed with double-distilled water and air-dried at RT up to the original moisture content (24 h).

An experimental greenhouse trial was conducted at the Department of Biology of the University of Rome Tor Vergata. Three growth cycles were managed from April 2022 to November 2023, using 2 pots for each treatment (6 pots per treatment at the end of the experiments).

The seeds were germinated in a towel (10 seeds each) soaked in 15 mL of double-distilled water and kept in the dark for 10 days at RT. Five germinated seeds were sown in plastic pots (15 cm diameter), containing about 350 g of soil (COMPO SANA® COMPACT, Münster, Germany). The soil characteristics were as follows: pH: 6.5; dry bulk density: 150 kg/m³; electrical conductivity (EC): 0.50 dS/m; porosity: 90% *v/v*. Soil components: neutral sphagnum peat, perlite (<5%), composted green soil improver. The plants (5 seedlings/pot) were grown in a greenhouse under natural sunlight (Daily Light Integral: 90 mmol photons/m² day ± 14 mmol photons/m² day), at a temperature of 26 °C ± 4 °C and soil moisture of 42% ± 6%; the environmental growth conditions were monitored daily using a multi-parameter sensor (Flower Care—HHCCJCY01HHCC—HHCC Plant Technology Co., Ltd.—Stuttgart, Germany).

After 14 days of growth, the primed and non-primed plants were watered with 100 mL of tap water or salt solution every 48 h for 4 weeks. Plants were irrigated with the following solutions: 20 mM NaCl (EC: 2.1 dS/m), 40 mM NaCl (EC: 4.2 dS/m), 80 mM NaCl (EC: 8.8 dS/m), and 160 mM NaCl (EC: 16.3 dS/m). The pots were randomly divided into the following experimental sets: (1) non-primed seedlings irrigated with tap water (EC: 0.6 dS/m) (control, CTRL) or with saline solutions; (2) primed seedlings irrigated either with tap water or with saline solutions. The pots' positions were changed at the time of watering.

### 2.3. Soil Analysis

The soil parameters were evaluated in pots with and without plants (blank) to estimate the uptake of minerals by the roots. At the end of the experiments, the gravimetric water content of the soil was determined according to Santangeli et al. [23]. The electrical conductivity (EC) of the soil was detected according to Sairam et al. [33] and measured with an EC meter (HANNA Instrument 98312 DiST®5 and DiST®6, Padova, Italy).

### 2.4. Morphological Parameters and Tolerance Index

Morphological parameters (length of the stem and the longest root, and the number of leaves) were evaluated at the end of the treatments. Harvesting was performed after 45 days of growth, and the tolerance index (TI) was determined according to Idrees et al. [34].

The plants were sampled (0.2 g fresh weight) and frozen by soaking in liquid nitrogen. The samples were kept at $-20\,^\circ$C until subsequent analyses.

### 2.5. Chlorophylls and Soluble Sugars

To quantify the chlorophyll content, frozen samples were prepared and processed as reported by Borromeo et al. [32]. The absorbances of the supernatants were evaluated by a spectrophotometer (VARIAN Cary 50 Bio, Santa Clara, CA, USA) at 664.1 nm (chlorophyll a) and 648.6 nm (chlorophyll b). The concentration of photosynthetic pigments was calculated according to Lichtenthaler [35] and expressed as $\mu$g·g f.w.$^{-1}$.

The quantification of monosaccharides was performed by the anthrone protocol, reported by Chun and Yin [36] with modifications described by Borromeo et al. [32]. Sample absorbance was measured at 625 nm with a spectrophotometer. The concentration of monosaccharides was calculated based on a calibration curve of glucose (y = 0.0121x + 0.0664; $R^2$ = 0.9947). The sugar content was expressed as mg glucose equivalent g f.w.$^{-1}$.

### 2.6. Quantification of Intracellular Free Calcium

Samples were prepared according to Borromeo et al. [32]. The calcium concentration was evaluated using the Calcium Assay Colorimetric Kit (Abcam, Cambridge, UK—ab272527; www.abcam.com/ab272527, accessed on 4 March 2024) and a multimode microplate reader set at 612 nm (Spark® Multimode Microplate Reader—Tecan, Switzerland). Data are expressed as $\mu$g Ca$^{2+}$·mg f.w.$^{-1}$.

### 2.7. Phenolic Compounds and Proline Content

Samples were prepared following the protocol reported by Borromeo et al. [32].

#### 2.7.1. Phenolic Compounds

The total phenolic content was determined using Folin–Ciocalteu reagent [23]. The absorbances of the standard and samples were measured at 724 nm by a spectrophotometer (VARIAN Cary 50 Bio). Phenolic compounds were estimated based on a calibration curve of chlorogenic acid (y = 0.0052x − 0.0231; $R^2$ = 0.9946). The total phenolic content was reported as $\mu$g chlorogenic acid equivalent·g f.w.$^{-1}$.

Flavonoids were assessed by detecting the absorbances at 415 nm with spectrophotometer [37]. Flavonoid concentration was determined with a calibration curve using quercetin as standard (y = 0.0067x − 0.0025; $R^2$ = 0.9982). Flavonoids were expressed as $\mu$g of quercetin equivalent·g f.w.$^{-1}$.

#### 2.7.2. Proline

Proline concentration was evaluated according to Stassinos et al. [29] by detecting the absorbance at 520 nm with a spectrophotometer (VARIAN Cary 50 Bio). The osmolyte level was calculated on the basis of a calibration curve of standard solutions of L-proline (y = 0.0684x − 0.0624; $R^2$ = 0.9963). Data were expressed as $\mu$g proline·g f.w.$^{-1}$.

## 2.8. Thiobarbituric Acid Reactive Products

Thiobarbituric acid (TBA) reactive products were estimated according to Micheli et al. [38] and Kaur and Jindal [39]. The sample absorbances (at 532 nm and 600 nm) were detected spectrophotometrically (VARIAN Cary 50 Bio). TBA reactive species were indicated as malondialdehyde (MDA) equivalent, according to the following formula:

$$\text{MDA equivalent (mmol/L)} = [(Abs_{532} - Abs_{600})/(\varepsilon \times l)]$$

where $\varepsilon$ = the extinction coefficient of MDA at 532 nm (155 mM$^{-1}\cdot$cm$^{-1}$); l = the path length of the cuvette (1 cm). Data were expressed as mmol MDA equivalent$\cdot$g f.w.$^{-1}$

## 2.9. Antioxidant Activity

Antioxidant activity, reducing power, and scavenger activity were measured utilizing the DPPH, PFRAP, and FRAP assays, respectively.

### 2.9.1. 2,2-Diphenyl-1-picryl-hydrazyl-hydrate (DPPH) Free Radical Assay

The percentage of antioxidant activity of each sample was assessed by the DPPH free radical assay and performed according to the protocol of Garcia et al. [40]. The half maximal inhibitory concentration (IC$_{50}$) of each analytical sample was calculated using the regression equations (y = ln(x)), where Y was replaced by 50. This value was set as the IC$_{50}$, expressed as mg/mL of each analytical sample.

### 2.9.2. Potassium Ferricyanide and Ferric Reducing Antioxidant Power (PFRAP and FRAP) Assays

Sample preparation for the FRAP and PFRAP assays required the same protocol: frozen samples were homogenized in 1.5 mL of methanol and were kept overnight at RT, then centrifuged at 4000× *g* for 15 min; supernatants were collected and stored at 4 °C until the analysis.

The PFRAP assay was performed according to Hue et al. [41]. The absorbance was recorded at 700 nm by a spectrophotometer; the scavenging activity was expressed as a %, compared to the untreated control (0 mM NaCl), set at 100% activity.

For the FRAP assay, based on the protocol of Gohari et al. [42] and Lim and Lim [43], the absorbance of the extracts was measured at 593 nm with the spectrophotometer. The ferric reducing power was expressed according to a calibration curve, made with known concentrations of FeSO$_4\cdot$7H$_2$O (y = 1.0818x − 0.0364; R$^2$ = 0.9974) and expressed as mmol FeSO$_4$ equivalent$\cdot$g f.w.$^{-1}$.

## 2.10. Enzymatic Activities

The enzymatic activities were associated with protein content in the sample analyzed. The protein concentration was determined by Bradford [44], using a calibration curve with bovine serum albumin (BSA) (y = 0.0372x + 0.0335; R$^2$ = 0.9946).

Superoxide dismutase (SOD) (EC 1.15.1.1) activity was evaluated by NPAGE (native polyacrylamide gel electrophoresis) [32]. SOD activity was visualized according to the procedure of Beauchamp and Fridovich [45]. The activity was expressed as arbitrary units (A.U.), corresponding to the pixel density of each lane obtained by the program Image J 1.53A.

Polyphenol oxidase (PPO) (EC 1.14.18.1) activity was determined according to Orzali et al. [46], with a few modifications [32]. The kinetics were followed by a spectrophotometer at the wavelength of 420 nm for 300 s. The activity of PPO was expressed as enzymatic units (E.U.$\cdot$mg protein$^{-1}$).

Peroxidase (POD) (EC 1.11.1.7) activity was investigated as described by Yang et al. [47] with modifications reported by Borromeo et al. [32]. The kinetics were determined by a spectrophotometer at a wavelength of 420 nm for 180 s. The enzyme activity was calculated as described by Yang et al. [47] and expressed as enzymatic units (E.U.$\cdot$mg protein$^{-1}$).

Ascorbate peroxidase (APX) (EC 1.11.1.11) activity was determined based on the method of Orzali et al. [46]. The rate of ascorbate oxidation was followed for 150 s by absorbance spectroscopy at a fixed λ of 290 nm (VARIAN Cary 50 Bio). Enzyme activity was expressed as a % compared to the untreated control (0 mM NaCl).

Catalase activity (CAT) (EC 1.11.1.6) was evaluated as reported by Iwase et al. [48] and calculated by measuring the height of the $O_2$ bubbles produced by the catalase. The activity was expressed as a %, compared to the untreated control (0 mM NaCl).

### 2.11. Fruit Characteristics

The bean reproductive phase started after 6 weeks, during which the treatments of the plants were prolonged, i.e., irrigation with saline and non-saline water, until all pods were fully ripened; after 2–3 weeks, the pods were harvested. Fruits were evaluated by the following parameters: (1) number of pods per plant, (2) pod weight and length, (3) number of seeds per pod, (4) weight of seeds per pod.

The seeds were stored at −20 °C and subsequently used for the quantification of proteins and soluble sugars. Finally, EC analysis of the soil was carried out at the end of the harvest to verify the final level of salinity.

### 2.12. Statistical Analysis

Data are reported as mean ± standard error (SE). One-way analysis of variance (ANOVA) was performed with Past 4.13. The Tukey–Kramer method was applied to determine the difference of significance among groups. All analyses were significant at $p < 0.05$ within each treatment group. When comparing primed groups to non-primed ones, the significance was *** $p < 0.001$; ** $p < 0.01$; * $p < 0.05$.

## 3. Results

### 3.1. Determination of Seeds' Halotolerance and Selection of Best Priming Agent

The halotolerance of seeds was assessed by a dose–response curve. All cultivars were sensitive or moderately sensitive to salt (Table 1) except for the cultivar Bola Roja, which was consequently excluded from the study.

**Table 1.** Germination rates of non-primed (CTRL) and primed (40 mM NaCl) seeds exposed to different salinity levels. Data are expressed as mean ± SE ($n = 6$). Significant differences within the same group ($p < 0.05$; ANOVA and Tukey–Kramer test) are reported with different letters in the column. Significant differences between primed and non-primed seeds are reported as * $p < 0.05$.

| Cultivar | NaCl (mM) | Non-Primed | Primed |
|---|---|---|---|
| Black Turtle | 0 | 53.2% ± 7.1% [a] | 63.8% ± 2.1% [a] |
| | 40 | 27.3% ± 3.3% [b] | 33.2% ± 3.0% [b] |
| | 80 | 32.1% ± 3.3% [b] | 31.9% ± 1.9% [b] |
| | 160 | 29.0% ± 5.4% [b] | 19.2% ± 2.8% [c*] |
| Bola Roja | 0 | 27.4% ± 6.6% [a] | 33.4% ± 3.3% [a] |
| | 40 | 29.1% ± 4.3% [a] | 32.7% ± 4.7% [a] |
| | 80 | 23.4% ± 3.7% [a] | 32.1% ± 4.1% [a] |
| | 160 | 28.1% ± 3.4% [a] | 33.1% ± 3.9% [a] |
| Borlotto | 0 | 73.1% ± 3.3% [a] | 69.3% ± 4.0% [a] |
| | 40 | 67.4% ± 6.8% [a] | 64.8% ± 2.1% [a] |
| | 80 | 40.1% ± 6.2% [b] | 43.1% ± 4.6% [b] |
| | 160 | 0% [c] | 3.0% ± 1.3% [c] |
| Cargamanto | 0 | 27.7% ± 2.9% [a] | 27.8% ± 2.2% [a] |
| | 40 | 13.8% ± 3.2% [b] | 23.1% ± 2.1% [ab*] |
| | 80 | 11.1% ± 2.2% [b] | 19.2% ± 2.9% [b] |
| | 160 | 6.2% ± 1.6% [c] | 8.4% ± 3.4% [c] |

According to the first results, the best priming agent was 40 mM NaCl (halopriming). After the priming treatment, halotolerance was different depending on the cultivar considered (Table 1). In cv. Borlotto, a significant decrease in the germination rate was observed with the increase in NaCl concentrations in the solution. A decrease of −33% in the rate was observed at 80 mM NaCl with respect to the control 0 mM NaCl, while germination was inhibited at 160 mM NaCl (Table 1). In the primed seeds, the reduction was −26% at 80 mM NaCl.

After the germination tests, the cultivar Borlotto was chosen and subjected to salt treatment since it fulfilled the following parameters: salt sensitivity, especially at a higher level of salinity; the absence of toxicity following the priming treatment; a high percentage of seed germination (both the primed and non-primed seeds); and economic and commercial importance. For this cultivar, we set up experiments to detect the physiological response to progressive exposure to salinity.

### 3.2. Soil Experiments and Analyses

We tested plant response to saline soil (EC = 3.9 dS/m) after halopriming and we observed no improvement in the tolerance. Similar observations were made when only acclimation to salt was performed. Similarly to results obtained with tomato [32], the single treatment was not sufficient to improve salt tolerance in bean plants. Thus, it was decided to use the same strategy applied to tomato plants: bean seeds, primed and non-primed, were sown in non-saline soil, acclimated for 2 weeks, and subjected to salt stress by irrigation for the following 4 weeks. Unexpectedly, the threshold value for salt tolerance, detected in the seed germination test (Table 1), did not correspond to the value detected during the following development period, being much lower. Indeed, irrigation with 80 mM and 160 mM NaCl caused severe damage (Figure 2a,b) resulting in plant death and confirming the high sensitivity of beans to salinity. Consequently, the plants were irrigated with 20 mM NaCl and 40 mM NaCl solutions. Gradual irrigation with such solutions allowed the plants to grow.

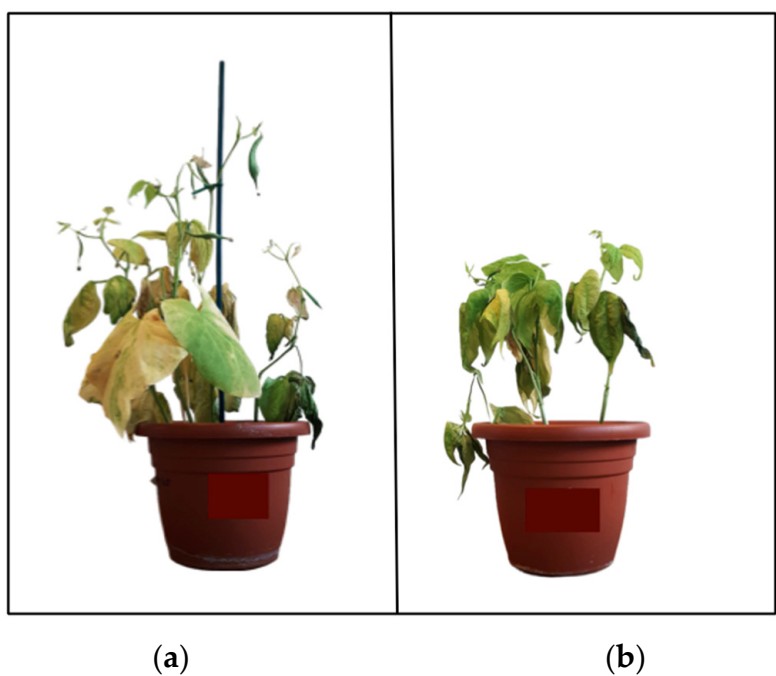

(**a**)         (**b**)

**Figure 2.** Bean plants (CTRLs) after 4 weeks of irrigation with 80 mM NaCl (**a**) and 160 mM NaCl (**b**).

At the end of the treatment, the gravimetric water content (GWC) and electrical conductivity (EC) of the soils were evaluated. The pots containing the primed plants showed a significantly lower GWC than the controls ($-61\%$ and $-50\%$ with 20 mM and 40 mM NaCl, respectively). These data were also confirmed by the analysis of EC, which was found to be lower in the pots irrigated with saline solutions ($-185\%$ and $-60\%$ with 20 mM and 40 mM NaCl, respectively) compared to the controls (Table 2).

**Table 2.** Gravimetric water content (GWC) and electrical conductivity (EC) of the soils at the end of the experiments. Each value represents mean $\pm$ SE ($n = 6$). Mean values in the column marked by different letters are significantly different within the same group ($p < 0.05$; ANOVA and Tukey–Kramer test). Significant differences to control (CTRL) are reported as * $p < 0.05$; *** $p < 0.001$.

| Priming Solutions | NaCl (mM) | Gravimetric Water Content (%) | Electrical Conductivity (dS/m) |
|---|---|---|---|
| CTRL | 0 | 1.28 $\pm$ 0.51 [a] | 0.246 $\pm$ 0.039 [a] |
| | 20 | 2.56 $\pm$ 0.10 [a] | 0.983 $\pm$ 0.085 [b] |
| | 40 | 4.23 $\pm$ 0.18 [b] | 1.596 $\pm$ 0.077 [c] |
| 40 mM NaCl | 0 | 1.16 $\pm$ 0.28 [a] | 0.338 $\pm$ 0.012 [a] |
| | 20 | 1.59 $\pm$ 0.03 [a***] | 0.345 $\pm$ 0.020 [a***] |
| | 40 | 2.82 $\pm$ 0.29 [b*] | 1.000 $\pm$ 0.073 [b***] |

*3.3. Effect of Saltwater Stress on Plant Growth*

In stressed non-primed plants (CTRLs), a decrease in shoot length was observed related to the increase in the salinity, while in the primed ones, a hormetic effect of priming was observed in the shoots. Moreover, these plants developed more leaves when compared to the corresponding CTRLs (Table 3), where the leaves showed chlorotic areas (Figure 3). The roots were longer in the primed plants, although not significantly different. The improved growth of the primed plants was confirmed by the tolerance index (TI), where a decrease in tolerance ($-12\%$ and $-22\%$) was recorded in the non-primed plants irrigated with 20 mM NaCl and 40 mM NaCl, respectively (Table 3).

**Table 3.** Morphological parameters and stress tolerance index (TI) of beans grown under different saline conditions. Data are reported as mean $\pm$ SE ($n = 7$). Mean values in the column marked by different letters are significantly different within the same group ($p < 0.05$; ANOVA and Tukey–Kramer test). Significant differences to CTRL are reported as ** $p < 0.01$; *** $p < 0.001$.

| Priming Solutions | NaCl (mM) | n. Leaves | Shoot Length (cm) | Root Length (cm) | TI (%) |
|---|---|---|---|---|---|
| CTRL | 0 | 20 $\pm$ 1 [a] | 46 $\pm$ 3.3 [a] | 21 $\pm$ 1.7 [a] | 100 |
| | 20 | 15 $\pm$ 2 [a] | 33 $\pm$ 3.7 [b] | 19 $\pm$ 3.0 [a] | 88 |
| | 40 | 15 $\pm$ 2 [a] | 38 $\pm$ 3.0 [ab] | 16 $\pm$ 0.8 [a] | 78 |
| 40 mM NaCl | 0 | 26 $\pm$ 1 [a**] | 63 $\pm$ 3.2 [a**] | 25 $\pm$ 0.4 [a] | 119 (+19%) |
| | 20 | 25 $\pm$ 2 [a**] | 64 $\pm$ 3.1 [a***] | 20 $\pm$ 1.5 [a] | 96 (+8) |
| | 40 | 19 $\pm$ 3 [a] | 60 $\pm$ 3.9 [a**] | 20 $\pm$ 2.2 [a] | 95 (+17) |

*3.4. Effect of Salt Treatment on Chlorophylls and Soluble Sugars*

In non-primed plants, little chlorosis was already observed at 20 mM NaCl, and wider chlorotic areas were detected with higher salt concentrations in the irrigation water (Figure 3), corresponding to the significant quantitative decrease in the chlorophylls (Table 4). Meanwhile, a slight increase in the level of soluble sugars was recorded after the irrigation with 40 mM NaCl (+11.2%) (Table 4).

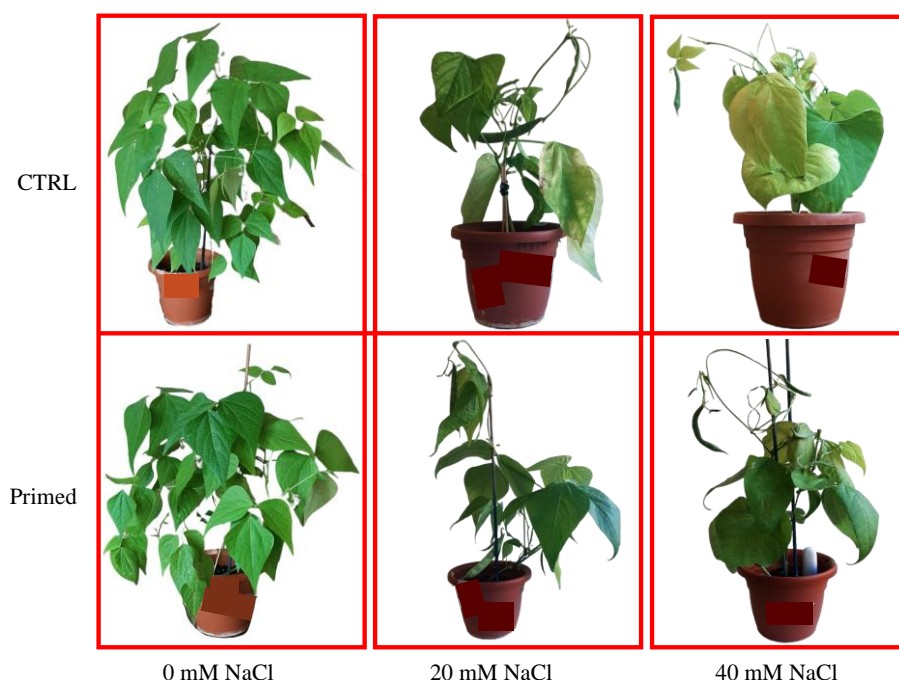

CTRL

Primed

0 mM NaCl 20 mM NaCl 40 mM NaCl

**Figure 3.** Primed and non-primed (CTRL) bean plants, irrigated with different saline solutions (0 mM NaCl, 20 mM NaCl, 40 mM NaCl), at the end of the experiments.

**Table 4.** Chlorophyll a (Chl a), chlorophyll b (Chl b), total chlorophyll content (total Chl), and soluble sugars of beans exposed to different saline conditions. Data are expressed as means $\pm$ SE ($n$ = 6). Mean values in the column marked by different letters are significantly different within the same group ($p < 0.05$; ANOVA and Tukey–Kramer test). Significant differences between groups are reported as * $p < 0.05$; *** $p < 0.001$.

| Priming Solution | NaCl (mM) | Chl a (µg·g f.w.$^{-1}$) | Chl b (µg·g f.w.$^{-1}$) | Total Chl (µg·g f.w.$^{-1}$) | Soluble Sugars (mg glucose eq.·g f.w.$^{-1}$) |
|---|---|---|---|---|---|
| CTRL | 0 | 142.8 $\pm$ 5.8 [a] | 85.5 $\pm$ 7.2 [a] | 228.3 $\pm$ 4.6 [a] | 0.250 $\pm$ 0.008 [ab] |
| | 20 | 96.7 $\pm$ 5.8 [b] | 56 $\pm$ 6.8 [b] | 152.7 $\pm$ 5.3 [b] | 0.229 $\pm$ 0.005 [a] |
| | 40 | 52.6 $\pm$ 5.1 [c] | 39.1 $\pm$ 8.9 [b] | 91.7 $\pm$ 5.7 [c] | 0.278 $\pm$ 0.010 [b] |
| 40 mM NaCl | 0 | 101.9 $\pm$ 2.5 [a***] | 43.7 $\pm$ 4.6 [a***] | 145.6 $\pm$ 4.1 [a***] | 0.203 $\pm$ 0.005 [a***] |
| | 20 | 112.2 $\pm$ 2.4 [a*] | 63.7 $\pm$ 11.2 [ab] | 175.9 $\pm$ 9.4 [b*] | 0.203 $\pm$ 0.003 [a***] |
| | 40 | 155.7 $\pm$ 4.4 [b***] | 71.3 $\pm$ 7.0 [b*] | 227 $\pm$ 3.3 [c***] | 0.238 $\pm$ 0.013 [b*] |

No significant damage was observed in the stressed primed plants (Figure 3), where chlorophylls significantly improved after saline irrigation with 40 mM NaCl (+53% of Chl a, +61% of Chl b, and +55% of total chlorophylls) (Table 4). The amount of soluble sugars was found to be significantly lower in the primed plants compared to the corresponding CTRLs (Table 4).

### 3.5. Calcium Translocation and Activation of the Response to Saltwater Stress

Calcium plays multiple roles in plants; under salt stress conditions, it works as a second messenger and activates signaling, leading to the onset of tolerance responses to stress [49]. Different behavior was detected in the shoots and roots of the primed and non-primed plants. In the latter exposed to salt, Ca$^{2+}$ increased in the roots, whereas a significant decrease was recorded in the shoots (Table 5).

**Table 5.** Calcium level in shoots and roots, reported as mean $\pm$ SE ($n$ = 6). Mean values in the column marked by different letters are significantly different within the same group ($p < 0.05$; ANOVA and Tukey–Kramer test). Significant differences to CTRL are reported as * $p < 0.05$; ** $p < 0.01$; *** $p < 0.001$.

| Priming Solution | NaCl (mM) | Shoot ($\mu$g Ca$^{2+}\cdot$mg f.w.$^{-1}$) | Root ($\mu$g Ca$^{2+}\cdot$mg f.w.$^{-1}$) |
|---|---|---|---|
| CTRL | 0 | 1.31 $\pm$ 0.03 [a] | 0.29 $\pm$ 0.02 [a] |
| | 20 | 0.71 $\pm$ 0.09 [b] | 0.38 $\pm$ 0.03 [b] |
| | 40 | 0.55 $\pm$ 0.06 [b] | 0.31 $\pm$ 0.03 [ab] |
| 40 mM NaCl | 0 | 1.09 $\pm$ 0.05 [a**] | 0.36 $\pm$ 0.03 [a] |
| | 20 | 1.74 $\pm$ 0.03 [b***] | 0.29 $\pm$ 0.01 [b**] |
| | 40 | 1.78 $\pm$ 0.04 [b***] | 0.24 $\pm$ 0.02 [b*] |

An opposite trend was observed in the stems and roots of the primed plants: the shoots had a higher Ca$^{2+}$ level, related to the salt concentration, while in the roots, a reduction in the Ca$^{2+}$ concentration was detected with the increase in salinity of the solutions (Table 5). These results suggest a rapid translocation of Ca$^{2+}$ from the roots to the stem of stressed primed plants associated with the activation of salt tolerance responses.

### 3.6. Changes in Metabolism: Phenolic Compounds and Proline

The increasing salinity of the irrigation water led to an imbalance in the synthesis of total phenols and flavonoids, as well as proline. In the CTRLs, saltwater led to a significant decrease in the production of total phenols ($-27.3\%$ and $-45.5\%$ with 20 mM and 40 mM NaCl, respectively) (Figure 4a) and flavonoids ($-50\%$ and $-56.3\%$ with 20 mM and 40 mM NaCl, respectively) (Figure 4b); additionally, a marked decrease in proline synthesis was observed ($-41\%$ and $-43.6\%$ at 20 mM and 40 mM NaCl, respectively) (Figure 4c). On the contrary, no significant reduction in the amount of phenols (Figure 4a), flavonoids (Figure 4b), and proline (Figure 4c) was observed in the stressed primed plants. A higher content of phenolic compounds was found in the stressed primed plants compared to the non-primed ones ($+12.5\%$ with 20 mM and $+50\%$ with 40 mM NaCl). The same pattern was observed for flavonoids ($+37.5\%$ and $+43\%$ in the primed stressed plants compared to the corresponding non-primed ones).

### 3.7. Lipid Peroxidation Inhibition

Thiobarbituric acid (TBA) reactive species were used as a marker for the determination of the degree of membrane lipid peroxidation, expressed as mmol malondialdehyde (MDA) eq./g f.w. A high amount of MDA suggests a high level of lipid peroxidation, resulting in damage to the plasma membrane. Salt exposure enhanced lipid peroxidation in the non-primed plants, thus damaging the membrane, while halopriming decreased this process ($-25\%$ with 40 mM NaCl compared to the corresponding CTRL), protecting the cells from the damage caused by saline irrigation (Figure 5).

### 3.8. Total Antioxidant Activity

Antioxidant activity, reducing power, and scavenger activity were assessed by the DPPH, FRAP, and P-FRAP tests, respectively. In the non-primed plants, a decrease in the antioxidant and scavenger activity was observed (Figures 6a–d and 7b), while no significant decrease in the reducing power was reported (Figure 7a). The primed stressed plants showed considerable increases in the antioxidant activity and reducing power (Figures 6a–d and 7a), as well as scavenger activity (Figure 7b), providing a better tolerance to salinity.

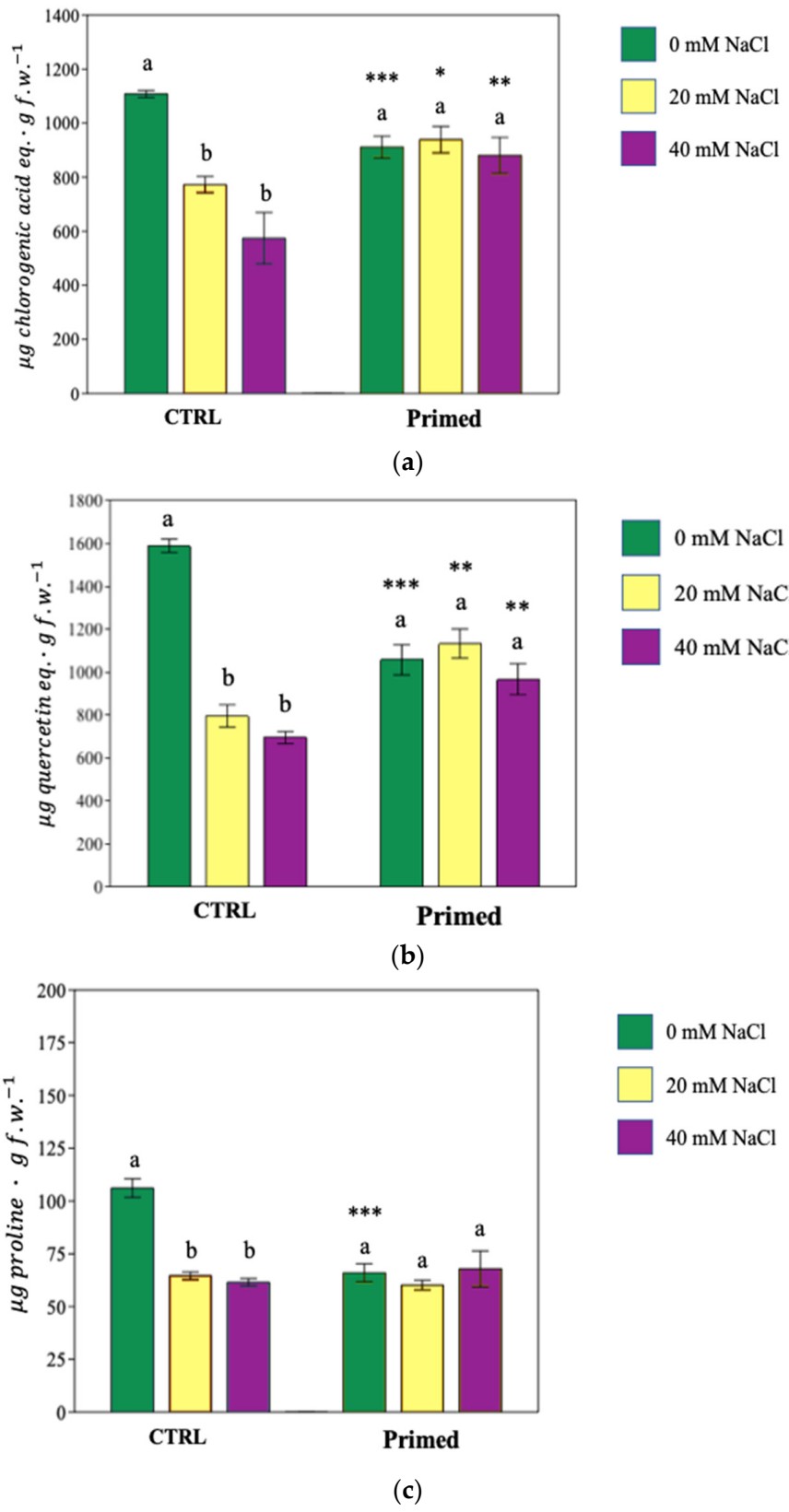

**Figure 4.** Phenolic compounds (**a**), flavonoids (**b**), and proline (**c**) in the plants. Data are expressed as mean $\pm$ SE ($n = 6$). Significant differences within the same group ($p < 0.05$; ANOVA and Tukey–Kramer test) are reported with different letters in the column. Significant differences to CTRL are reported as * $p < 0.05$; ** $p < 0.01$; *** $p < 0.001$.

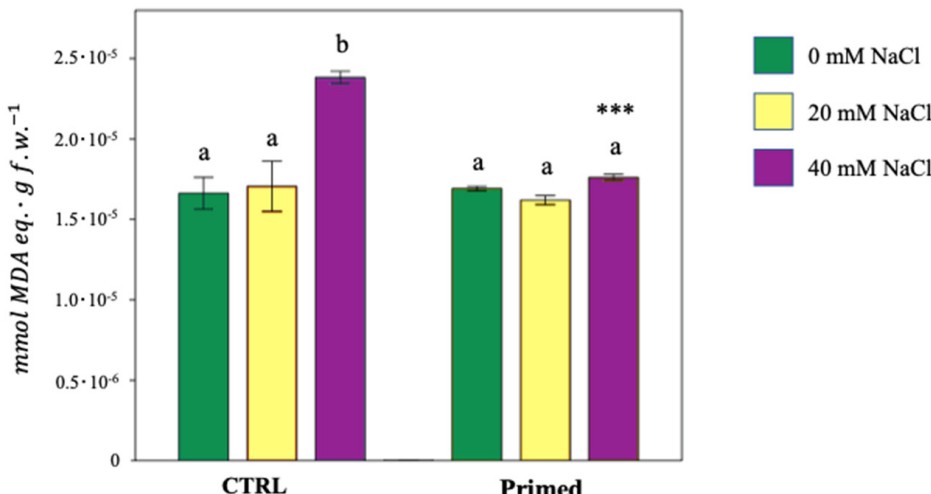

**Figure 5.** Thiobarbituric acid reactive products in bean samples. Data are expressed as mean ± SE (*n* = 6). Significant differences within the same group ($p < 0.05$; ANOVA and Tukey–Kramer test) are reported with different letters in the column, while significant differences to CTRL are reported as *** $p < 0.001$.

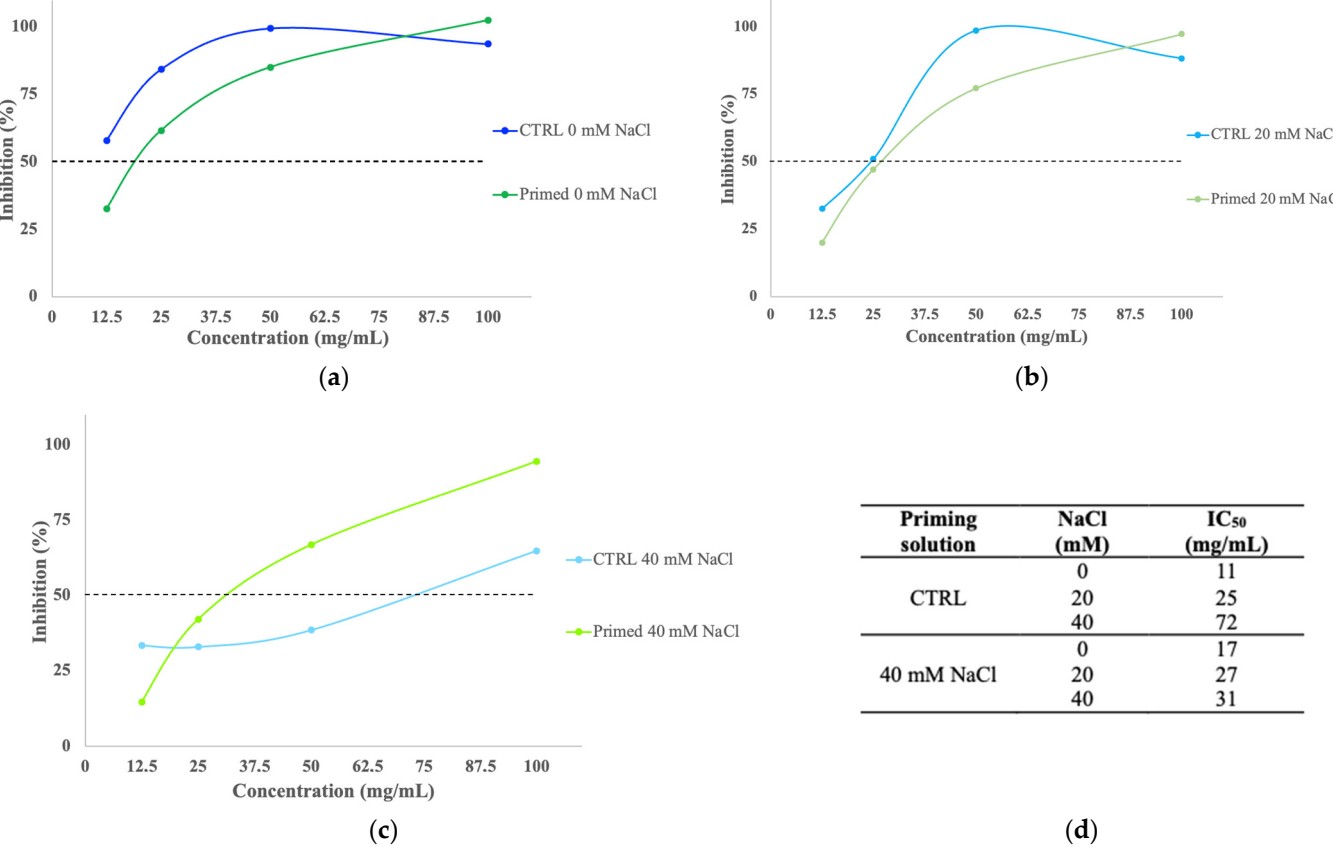

**Figure 6.** DPPH free radical inhibiting activity (%) at various concentrations of leaf extract (mg/mL) (**a**–**c**) and estimation of $IC_{50}$ value of plants (**d**). The dotted line represents 50% inhibition.

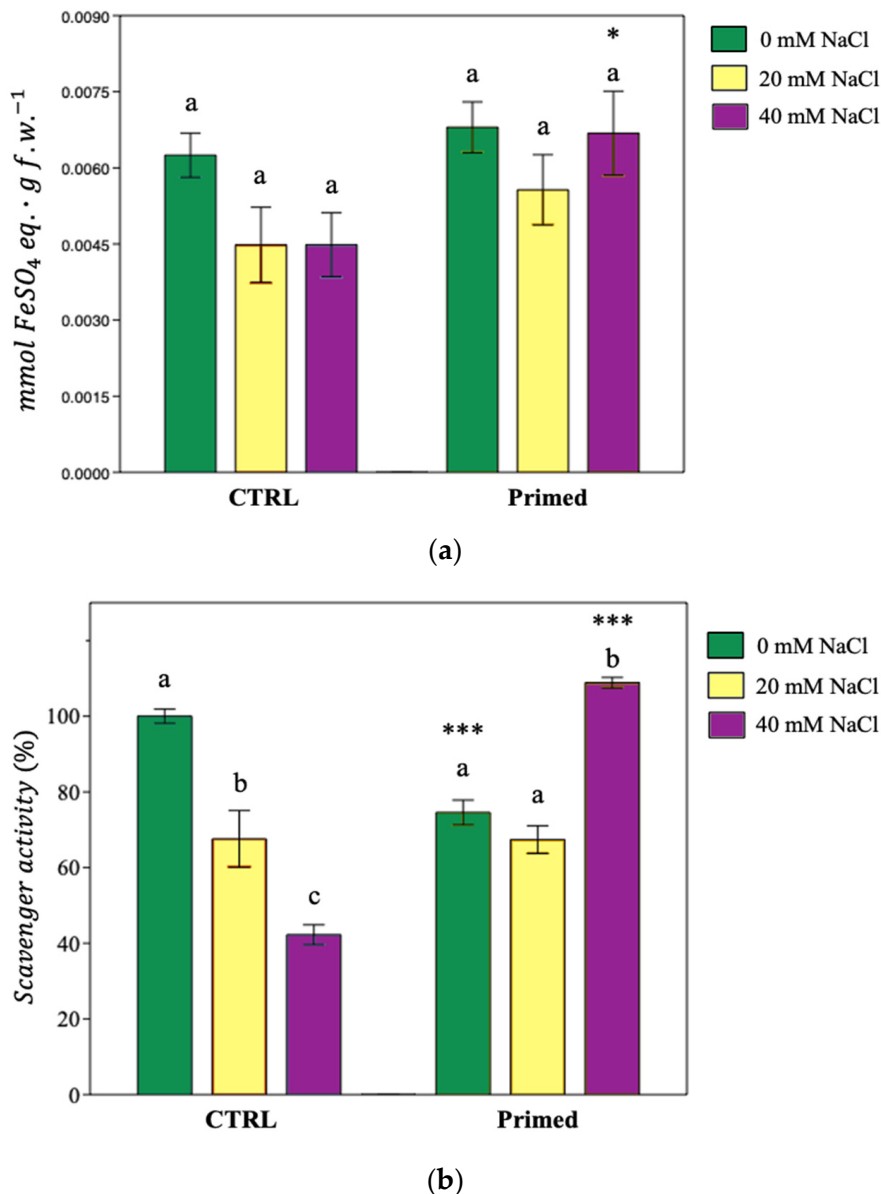

**Figure 7.** Ferric reducing antioxidant power (**a**) and scavenging activity (**b**) of plants. Results are expressed as mean $\pm$ SE ($n$ = 6). Significant differences within the same group ($p < 0.05$; ANOVA and Tukey–Kramer test) are reported with different letters in the column, while significant differences to CTRL are reported as * $p < 0.05$; *** $p < 0.001$.

### 3.9. Enzymatic Activities

The activity of various antioxidant enzymes was evaluated. Superoxide dismutase (SOD) was more active in both the primed and non-primed plants under salt stress (Figure 8a), showing significantly higher activity (+63%) in the primed plants irrigated with 40 mM NaCl compared to the CTRLs (Figure 8b).

In the non-primed plants, the activity of peroxidases (PODs), particularly ascorbate peroxidase (APX), was reduced following salt stress (Table 6); also, PPO and CAT (Table 6) showed a similar response. An opposite pattern was found in the primed stressed plants, where an improvement in POD, CAT, and APX activities was recorded after saline irrigation (Table 6). Instead, PPO activity improved only under high salt stress (40 mM NaCl) (Table 6).

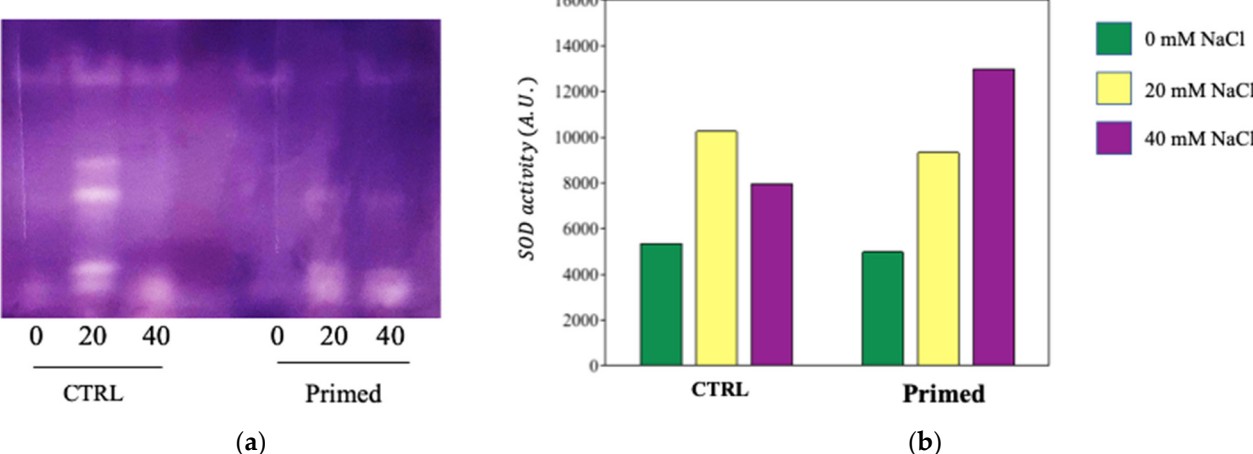

**Figure 8.** Native gel of bean plant extracts (**a**) and SOD activity of bean plants expressed as arbitrary units obtained by the program Image J 1.53a (**b**).

**Table 6.** PPO, POD, APX, and CAT activities in plants under different saline conditions. Enzyme activities are expressed as % compared to the untreated CTRL (0 mM NaCl). Data are expressed as mean $\pm$ SE ($n = 3$). Significant differences within the same group ($p < 0.05$; ANOVA and Tukey–Kramer test) are reported with different letters in the column. Significant differences to CTRL are reported as * $p < 0.05$; ** $p < 0.01$; *** $p < 0.001$.

| Priming Solution | NaCl (mM) | PPO Activity (E.U.·mg protein$^{-1}$) | POD Activity (E.U.·mg protein$^{-1}$) | APX Activity (%) | CAT Activity (%) |
|---|---|---|---|---|---|
| CTRL | 0 | 0.108 $\pm$ 0.007 [a] | 0.186 $\pm$ 0.001 [a] | 100 [a] | 100 [a] |
| | 20 | 0.120 $\pm$ 0.019 [a] | 0.153 $\pm$ 0.010 [ab] | 82 $\pm$ 6 [a] | 98 $\pm$ 10 [a] |
| | 40 | 0.062 $\pm$ 0.003 [b] | 0.112 $\pm$ 0.017 [b] | 20 $\pm$ 11 [b] | 70 $\pm$ 10 [a] |
| 40 mM NaCl | 0 | 0.022 $\pm$ 0.001 [a**] | 0.269 $\pm$ 0.070 [a] | 50 $\pm$ 7 [a**] | 70 $\pm$ 10 [a] |
| | 20 | 0.021 $\pm$ 0.006 [a***] | 0.433 $\pm$ 0.069 [a*] | 91 $\pm$ 6 [b] | 134 $\pm$ 16 [b] |
| | 40 | 0.191 $\pm$ 0.014 [b***] | 1.194 $\pm$ 0.176 [b***] | 93 $\pm$ 11 [b***] | 144 $\pm$ 12 [b**] |

### *3.10. Effect of Salt Treatment on Pod Production and Seed Analysis*

The priming treatment led to a general improvement in plant productivity. In the non-primed plants irrigated with saltwater, a decline in pod development and ripening was observed (Figure 9), leading to a decrease in both pod and seed weight (Table 7). Moreover, the seed morphology was also affected by salinity (Figure 10): some seeds showed abnormal coloration and poor ripening, as well as reduced size, particularly after 40 mM NaCl irrigation.

Priming counteracted the effects of salt stress, leading to an increase in production, weight, and pod length (Table 7), particularly at lower salt irrigation. Moreover, the most significant increase in seed weight was observed in the primed stressed plants: the treatment improved their weight by factors of 3 and 9 with 20 and 40 mM NaCl, respectively (Table 7).

The amount of proteins and soluble sugars was also found to be significantly increased in seeds obtained from the primed plants (Table 8), particularly following irrigation with 40 mM NaCl (by a factor of 4 and 11 for proteins and sugars, respectively), while a marked decrease in both proteins and sugars was detected in the seeds of the CTRLs, related to the increasing salinity. No significant changes in EC were recorded in the soils of the primed and non-primed plants (Table 8).

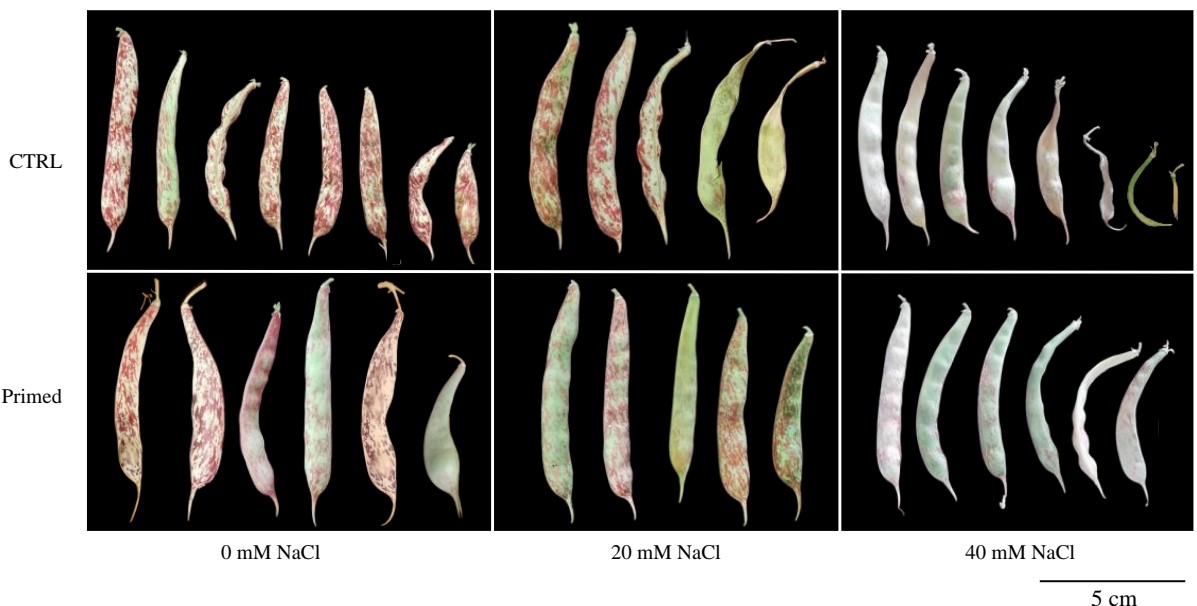

**Figure 9.** Pods yielded from primed and non-primed plants, exposed or not exposed to salt.

**Table 7.** Morphological parameters of pods and seeds collected from plants exposed to different saline conditions. Data are expressed as mean values $\pm$ SE (*n* = 10). Significant differences within the same group (*p* < 0.05; ANOVA and Tukey–Kramer test) are reported with different letters in the column, while significant differences to control (CTRL) are reported as * *p* < 0.05; ** *p* < 0.01.

| Priming Solution | NaCl (mM) | n. Pods/Plant | Pod Weight (g) | Pod Length (cm) | n. Beans/Pod | Bean Weight/Pod (g) |
|---|---|---|---|---|---|---|
| CTRL | 0 | 1.5 ± 0.3 [a] | 3.5 ± 0.3 [a] | 10 ± 0.4 [a] | 1.8 ± 0.3 [a] | 0.778 ± 0.052 [a] |
| | 20 | 1.6 ± 0.3 [a] | 2.5 ± 0.3 [a] | 9 ± 0.4 [a] | 1.5 ± 0.2 [a] | 0.092 ± 0.012 [b] |
| | 40 | 2.1 ± 0.3 [a] | 1.4 ± 0.3 [b] | 8 ± 0.3 [a] | 2.1 ± 0.2 [a] | 0.023 ± 0.005 [c] |
| 40 mM NaCl | 0 | 1.8 ± 0.3 [a] | 5.0 ± 0.4 [a*] | 10 ± 0.4 [a] | 2.2 ± 0.2 [a] | 0.959 ± 0.067 [a] |
| | 20 | 2.9 ± 0.4 [b*] | 2.9 ± 0.3 [b] | 9 ± 0.5 [a] | 1.8 ± 0.3 [a] | 0.254 ± 0.050 [b*] |
| | 40 | 2.2 ± 0.2 [ab] | 2.9 ± 0.3 [b*] | 10 ± 0.3 [a**] | 2.5 ± 0.4 [a] | 0.205 ± 0.037 [b**] |

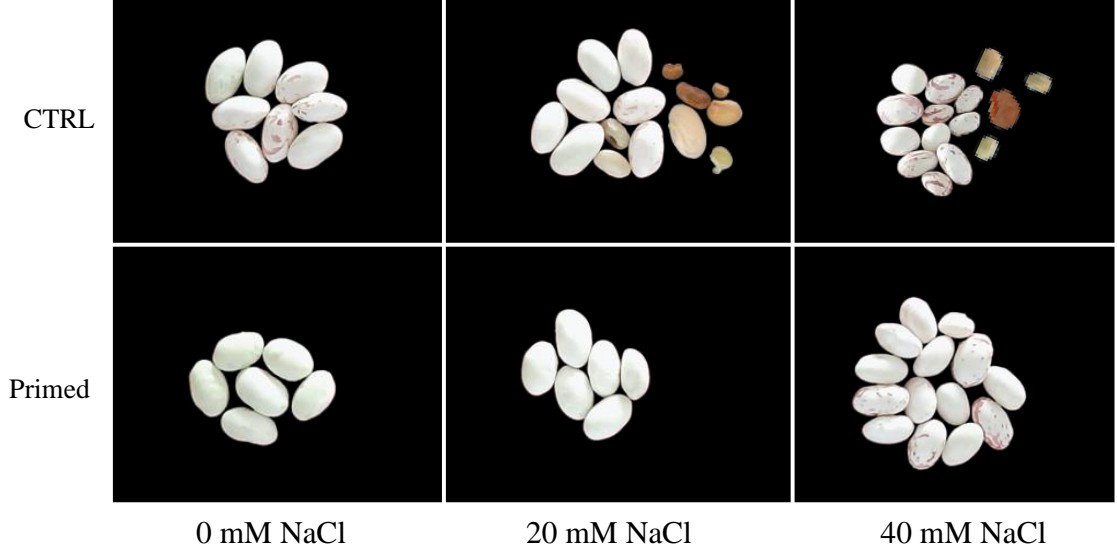

**Figure 10.** Seeds of bean plants exposed to different salt solutions (0 mM, 20 mM, and 40 mM NaCl).

**Table 8.** EC of soil at the end of harvest and protein and soluble sugar concentration in seeds collected from plants exposed to different salt solutions. Data are expressed as mean $\pm$ SE ($n$ = 3). Mean values in the column marked by different letters are significantly different within the same group ($p < 0.05$; ANOVA and Tukey–Kramer test). Significant differences to CTRL are reported as * $p < 0.05$; ** $p < 0.01$; *** $p < 0.001$.

| Priming Solutions | NaCl (mM) | Electrical Conductivity of Soils (dS·m$^{-1}$) | Proteins of Seeds (µg BSA eq.·mg f.w.$^{-1}$) | Soluble Sugars of Seeds (mg glucose eq.·g f.w.$^{-1}$) |
|---|---|---|---|---|
| CTRL | 0 | 0.243 $\pm$ 0.018 [a] | 6.31 $\pm$ 0.26 [a] | 0.93 $\pm$ 0.10 [a] |
| | 20 | 0.573 $\pm$ 0.061 [b] | 6.18 $\pm$ 0.79 [a] | 0.43 $\pm$ 0.12 [b] |
| | 40 | 0.600 $\pm$ 0.076 [b] | 1.67 $\pm$ 0.13 [b] | 0.08 $\pm$ 0.02 [c] |
| 40 mM NaCl | 0 | 0.290 $\pm$ 0.021 [a] | 7.12 $\pm$ 0.08 [a*] | 1.19 $\pm$ 0.01 [a*] |
| | 20 | 0.637 $\pm$ 0.072 [b] | 6.80 $\pm$ 0.05 [a] | 0.86 $\pm$ 0.02 [b**] |
| | 40 | 0.773 $\pm$ 0.055 [b] | 7.04 $\pm$ 0.21 [a***] | 0.84 $\pm$ 0.13 [b***] |

## 4. Discussion

Climate change causes concern to the scientific community due to the negative impacts on crop production worldwide. Climate change scenarios include variations in rainfall and extreme weather events [50,51]. The common bean is one of the crop species most threatened by this change since it is cultivated in regions subject to strong variations in environmental conditions (e.g., Africa and Central and South America). This species is sensitive to a wide range of abiotic stresses, such as drought and salinity [52–54].

Salinity stress, induced by salt accumulation in soil, negatively affects agricultural sustainability and productivity by reducing the seed germination, growth, and physiological characteristics of plants, resulting in limited crop production and yield [32,55]. According to the FAO (Food and Agriculture Organization, https://www.fao.org/3/y4263e/y4263e0e.html (accessed on 10 December 2023), the common bean is considered one of the most salt-sensitive species, together with carrot (*Daucus carota* L.), onion (*Allium cepa* L.), and strawberry (*Fragaria x ananassa* Duch.).

Since beans are an important crop in the world, it is essential to protect their cultivation and yield, even under salt stress conditions [56,57]. An enhancement in tolerance to salt stress can be achieved by various approaches [23,29,31]. In this work, we used an environmentally friendly and cost-effective protocol, known as seed priming, which involves the exposure during the pre-germination phase of seeds to elicitors (i.e., NaCl) that induce a slight stress condition. Primed seeds will maintain so-called "stress memory" and will perform better than unprimed ones when they are subsequently exposed to a stress [29,32].

### 4.1. Effect of Seed Priming on Bean Germination and Growth

We investigated the impact of priming on bean cultivars exposed to different saline and non-saline conditions. Interestingly, different effects were observed among the cultivars, e.g., in Black Turtle, a decrease in the germination of the primed seeds compared to the non-primed ones was observed. On the other hand, no significant changes in germination were found in Bola Roja, while in Cargamanto, germination increased only under low salinity. Our results support the theory, reported by many authors [58–61], that salt response after seed priming is species- or even cultivar-specific and there is no universally effective type of treatment to increase plant tolerance to salt.

A decrease in germination was observed in Borlotto, where the rate was very low at 160 mM. Considering the economic importance of this cultivar, we focused the next experiment on the physiological response of Borlotto, starting from sowing germinated seeds, either primed or not, in saline soil. Despite reports from the literature [62–65], priming was not sufficient to support plant growth in such conditions. This outcome had already been observed in our previous study on tomato plants [32], where an improvement in salt tolerance was observed when priming was followed by an acclimation phase to

salinity. Thus, we wondered if beans could have the same behavior, i.e., if the halopriming would confer to the beans a certain salt tolerance in the later growth stage. Primed and non-primed bean seeds were sown in non-saline soil, irrigated for two weeks with non-saline water, and then subjected to different saline and non-saline irrigation regimes for the next four weeks of growth. Based on the high salt sensitivity of beans [13], we tested irrigation regimes at different NaCl concentrations. Salinity tolerance thresholds changed depending on the development stages; i.e., during germination in the dose–response curve study, we observed higher salt resistance, with a highest tolerable salt concentration of 80 mM NaCl. Subsequently, the tolerance level of the plants decreased in the seedling stage when the maximum tolerable concentration dropped to 40 mM NaCl. Above this threshold, the plants were so deeply affected that they died prematurely. Consequently, it was decided to irrigate the plants with 20 mM and 40 mM NaCl to allow plant growth and assess, in the meantime, the plant response to salinity.

Halopriming improved the growth in saline conditions; plants developed new leaves and had better stem growth. The enhanced vigor of the primed plants, compared to the non-primed controls, is proof of a higher tolerance index against salt stress observed in seedlings. These results confirm previous observations on rapeseed [66], wheat [67], and tomato [68] plants, where halopriming reduced the inhibitory effects of salinity on seedling establishment.

The positive role of priming was also played during the acclimation process, when the plantlets were able to counteract stress conditions. Moreover, soil analysis revealed that halopriming increased plant water uptake. The soil of the control plants showed a higher water content associated with a higher EC value, while lower values of both water content and EC were detected in the soil of the primed plants. These results agree with the observations by [69], where priming increased water uptake in rice subjected to salt stress.

*4.2. Relationship between Seed Priming, Photosynthesis, and Sugar Metabolism*

Even though controversy exists in the literature concerning the effect of salinity on photosynthesis, most authors agree that salt stress reduces the rate of photosynthesis [70–72]. This reduction can be caused by the degradation or instability of chlorophylls, by the inhibition of enzymes necessary for their synthesis, or, according to other authors, a destruction of the chloroplast caused by salinity [17,32,73,74]. We observed a decrease in chlorophyll amount in the non-primed stressed plants, while halopriming improved photosynthetic efficiency by decreasing the rate of chlorophyll degradation even under stress conditions, as shown by the high total chlorophyll content.

Changes in the amount of carbohydrates are considered important due to their involvement in various physiological processes, such as photosynthesis. Soluble sugars are considered osmolytes and key factors in osmotic regulation for resistance to various stresses [75,76]. In plants exposed to salt, an increase in sugar content was observed; this phenomenon contributes to osmotic regulation, helping to maintain basal plant metabolism even under adverse conditions [32,75]. Similarly to what was reported in these studies, in our experiments, the amount of these osmolytes in the leaf tissues of non-primed stressed plants was higher than in primed ones, supporting the idea that priming is an efficient tool to balance salt-induced osmotic stress.

*4.3. Activation of Signaling and Salt Tolerance Responses*

In plants grown in saline soils, salt accumulation causes ionic imbalance that results in ionic stress, which inhibits the translocation of various ions, such as $K^+$, $Ca^{2+}$, and $Mg^{2+}$, to the stem and leaves, preventing the growth of young plants [16,23,29]. The activation of adaptation responses towards this stress is mediated by calcium [49]. However, excessive salt in the roots inhibits calcium translocation to the shoot, delaying the activation of adaptive responses and, in severe cases, causing plant death [32,49]. In our experiments, halopriming improved both calcium translocation and the activation of salt response; in the stem of the control plants, irrigated with salt solutions, the amount of calcium was

significantly lower than in the primed stressed plants. The opposite results, determined in the roots, support the idea that priming increased calcium translocation, leading to a more rapid activation of salt adaptive responses.

As a consequence of ionic and osmotic stresses, due to salinity, the overproduction of ROS results in oxidative damage to plasma membrane components and cell organelles, particularly the mitochondria, disrupting electron transport and promoting the release of reactive oxygen species [77]. These radicals cause lipid peroxidation. In agreement with these studies, an increase in MDA concentration was observed in the non-primed stressed plants, suggesting a high degree of membrane lipid peroxidation. Priming is known to better preserve plasma membrane [29] and mitochondrial integrity [77]. Our data confirm that halopriming reduced the level of MDA, enhancing the integrity of the plasma membrane, under stress conditions. The enhancement in antioxidant defense, induced by halopriming, can be an effective tool to counteract salt-induced oxidative stress. In fact, an increase in scavenger activity and reducing power was detected in all primed stressed plants (particularly following irrigation with 40 mM NaCl). In agreement with Salami et al. [78], we found an accumulation of secondary metabolites, in particular flavonoids, observed in all primed and stressed plants, suggesting that halopriming leads to the activation of a non-enzymatic antioxidant response. Proline is involved in the response to salinity by protecting cells from oxidative damage and osmotic stress [79], thus enhancing plant growth under stress conditions; for this reason, proline is considered an important non-enzymatic antioxidant against stress-induced ROS and, therefore, a good indicator of salinity tolerance [79]. Data from the literature on the synthesis of proline in stress conditions are controversial. For example, an increase in proline has been observed in many studies on primed plants exposed to stress [29,71,72,79]; vice versa, some authors [80,81] found no change in the production of this osmolyte in stressed plants. In our study, no proline accumulation was observed in the primed stressed plants. The lack of accumulation of this osmolyte was compensated by a marked activation of other antioxidant systems, which was detected in all primed stressed plants. Indeed, a direct correlation between antioxidant enzymes and salt-induced tolerance to oxidative stress has been reported in several studies [14,30,46,77,81,82]. Following halopriming, a general enhancement in antioxidant enzymatic activities was found in plants exposed to salinity; in particular, POD, APX, and CAT exhibited significantly higher activities than the corresponding stressed controls (+966%, +365%, and +106% after irrigation with 40 mM NaCl). These results support the hypothesis that priming can activate antioxidant defense systems, either enzymatic or non-enzymatic, providing an optimal response against ROS overproduction caused by increasing salinity.

### 4.4. Seed Priming during Reproductive Phase: Effect on Pod Production and Seed Quality

The common bean is a crop with a relatively short growing period; i.e., pod and seed production occurs within 2–3 months after sowing. Even if the common bean is the most widely grown and marketed legume in the world [3], there are no studies focusing on the yield and nutritional properties of bean seeds produced by primed plants grown under stress. Therefore, a part of our work was dedicated to the evaluation of the effect of halopriming on bean reproduction. No difference was detected in the anthesis between non-primed and primed plants, but in the latter, a yield improvement was observed. An increase in the number of pods per plant (+50% with respect to the control) was determined in plants irrigated with 20 mM NaCl solution, while the number of seeds in a single pod remained unchanged. Moreover, halopriming increased both the length and the weight of the pods. An extremely significant improvement was found in the weight of seeds produced by primed plants irrigated with 40 mM NaCl (+791% compared to the control). While salinity reduces yield, there is evidence that, when the salt stress is moderate, it can have a positive impact on the quality parameters of fruit and vegetables [83]. However, as reported in the EIP-AGRI Minipaper [84], information about the nutritional quality of vegetables grown in salinized environments is still scarce.

Since the principal nutrient components of the common bean are sugars and proteins [2,3], we evaluated the amounts of proteins and sugars in seeds produced by stressed and non-stressed primed plants, and we found that the priming treatment improved the sugar and protein levels in these seeds.

As far as the EC of the soil is concerned, the irrigation with saltwater did not significantly increase the EC during the growing period: the maximum value was 1.6 dS/m, found in the soil of the non-primed stressed plants, with a $\Delta = 1.1$ dS/m after 6 weeks of irrigation with 40 mM NaCl. The soil EC values decreased during seed ripening, showing a similar trend in both the primed and non-primed plants. At the end of the experiments (harvesting period), the EC of the soil was not significantly different (the maximum value was 0.77 dS/m, detected in the soil of the primed plants, irrigated with 40 mM NaCl) from that of the soil at the beginning of the experiments (0.50 dS/m).

Based on these results, to cope with the depletion of water resources and the higher requirement of water for bean cultivation, the use of slightly saline water for irrigation can be envisaged. However, the long-term effects of salinization on chemical and physical soil parameters deserve further investigation.

## 5. Conclusions

Salt stress has a negative impact during all developmental stages of bean plants, from germination to reproduction. However, there are a limited number of works investigating the consequences of salt stress regarding the entire life cycle of the plant and, even less, those studying the effect of priming on stressed plants up to the pod production stage. Consequently, the aim of this work was to study the effect of seed priming and acclimation during the life cycle of bean plants subjected to salt stress. This work showed that seed priming and acclimation are useful tools to increase the salt tolerance of the bean, improving growth, water uptake, chlorophylls, and the concentration and activity of antioxidant molecules and enzymes, in turn reducing cell damage caused by prolonged salt irrigation. This technique also stimulated pod production and significantly increased the protein and sugar content of the seeds, making them more nutrient-rich than non-primed counterparts. In addition to promoting the plant's response and adaptation to stress conditions, halopriming and acclimation represent an easy and economical technique that can be applied by farmers to overcome the problem of salt stress in this glycophyte legume. Further research aiming to study the positive impact of priming should be supported to promote the wider use of this technique.

**Author Contributions:** Conceptualization, I.B., C.F. and M.D.G.; methodology, I.B. and C.F.; software, I.B. and F.D.; validation, I.B. and F.D.; formal analysis and investigation, I.B. and F.D.; data curation, I.B. and C.F.; writing—original draft preparation, I.B. and C.F.; writing—review and editing, I.B., C.G., M.D.G. and C.F.; project administration, C.F. All authors have read and agreed to the published version of the manuscript.

**Funding:** This research received no external funding.

**Institutional Review Board Statement:** Not applicable.

**Informed Consent Statement:** Not applicable.

**Data Availability Statement:** Data are contained within the article.

**Conflicts of Interest:** The authors declare no conflicts of interest.

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
