# Peer review of "Enhancing Bean (Phaseolus vulgaris L.) Resilience: Unveiling the Role of Halopriming against Saltwater Stress"

_2674-1024, doi:10.3390/seeds3020018_

Round 1

Reviewer 1 Report

Comments and Suggestions for Authors

Overall the manuscript is quite potet. However, some rectifications should be incorporated as mentioned in the attachment to make it scientifically more attactive. 

.

Comments on the Quality of English Language

Minor editing is fine.

Author Response

Reply to REFEREE 1

The authors wish to thank the reviewer for the work and good suggestions to which we reply point by point

An appreciable attempt has been made by the authors to study the indispensable role of halopriming against saltwater stress in view of enhancing bean resilience. However, the authors may go through some modifications as suggested below to improve the overall quality of the manuscript to be potent candidate for the concerned journal:

ABSTRACT

Abstract should be more well-organized and compact with certain distinction among all the chapters covered in the main manuscript. Unnecessary background study should be avoided. Methods should be described in a nutshell. Major findings should be mentioned with at least some numeric results. The pinpoint vision of this present literature should be mentioned more clearly in a single sentence along with the outcome emanated from the research at the end of the abstract for better understanding of the readers.

Reply: abstract was rewritten taking the suggestions into consideration (lines 17-30).

INTRODUCTION

The authors are suggested to include more number of recent literatures and to eliminate preferably the older ones as far as possible. There are several important facts mentioned in the introduction without specific literature citation. Why the authors are going for specifically this work should be cleared in the introduction. Authors are suggested to thoroughly check the introduction as well as the whole manuscript for minor typographical errors.

Reply: done (cit. 8, 16, 22, 28)

MATERIALS AND METHODS

Up to the mark.

RESULTS

Result section is quite satisfactory.

DISCUSSION

There is a serious lacking of substantial literature evidences in support of the research outcomes. Authors are suggested to carefully review this issue. Authors are suggested to divide this section into sub sections according to the parameters mentioned in results section for better understanding of the whole idea.

Reply: the discussion was revised and divided into paragraphs to make it easier to understand (line 513, 556, 575, 620)

CONCLUSION

The conclusion is the actual reflection of the whole research which should be presented in a brief but structured manner. Try to conclude from the outcomes of the experiment and mention properly. Here this segment is too short and does not specify the actual result of the current experiment. Therefore, conclusion should be rephrased.

Reply: conclusion was rewritten (lines 656-671)

Reviewer 2 Report

Comments and Suggestions for Authors

The current manuscript was well designed and written. Before the manuscript can be further considered for publication, the following concerns should be addressed as raised by the reviewer.

1. The abstract section should be rewritten. Line 17-27 should be simplified to one or two sentences. It is suggested to state the limitations of the previous studies and strengthen the innovation of the present work. Line 28-36 should also be simplified. It is suggested that more detailed results should be stated here other than the experimental design. Although a lot of physiological parameters have been investigated in the whole experiment, very little results have been mentioned here.

2. In materials and methods section, they authors failed to describe the replications of the experiment. In figure legends, n=3 and n=6 have been observed. The authors should describe this information in more detail.

3. Apart from the pictures of the four cultivars (Figure 1), more information should be provided concerning these cultivars. It is suggested to describe the reason they use these cultivars.

4. In Table 1, the data of germination rates was confused. Usually germination rates should be described in "X%". Besides, no decimal points was observed in each digit.

5. In Table 2, the last digit "1" of EC should be "1.000".

6. In Figure 4 and 5, the titles of the vertical coordinates were missing. Only units were observed here.

7. Data of antioxidant enzymes should be combined. Figure 9. Figure 10, and Table 6 should be combined to one figure.

8. In discussion section, the potential relations among the parameters should be further discussed, not discuss them separately.

9. The conclusion section should be written. The authors should describe and strengthen their findings and major results of the present work, not the significance of the present work.

Author Response

Seeds 2936464 Reply to Referee 2

The authors wish to thank the reviewer for the work and good suggestions to which we reply point by point

The current manuscript was well designed and written. Before the manuscript can be further considered for publication, the following concerns should be addressed as raised by the reviewer.

  1. The abstract section should be rewritten. Line 17-27 should be simplified to one or two sentences. It is suggested to state the limitations of the previous studies and strengthen the innovation of the present work. Line 28-36 should also be simplified. It is suggested that more detailed results should be stated here other than the experimental design. Although a lot of physiological parameters have been investigated in the whole experiment, very little results have been mentioned here.

Reply: abstract was rewritten and simplified (line 17-30)

  1. In materials and methods section, they authors failed to describe the replications of the experiment. In figure legends, n=3 and n=6 have been observed. The authors should describe this information in more detail.

Reply: done (lines 123-126)

  1. Apart from the pictures of the four cultivars (Figure 1), more information should be provided concerning these cultivars. It is suggested to describe the reason they use these cultivars.

Reply: done (lines 84-87)

  1. In Table 1, the data of germination rates was confused. Usually germination rates should be described in "X%". Besides, no decimal points was observed in each digit.

Reply: done (line 295)

  1. In Table 2, the last digit "1" of EC should be "1.000".

Reply: done (line 329)

  1. In Figure 4 and 5, the titles of the vertical coordinates were missing. Only units were observed here.

Reply: the titles are visible in all graphs

  1. Data of antioxidant enzymes should be combined. Figure 9. Figure 10, and Table 6 should be combined to one figure.

Reply: done (line 453)

  1. In discussion section, the potential relations among the parameters should be further discussed, not discuss them separately.

Reply: the discussion was revised and divided into paragraphs to make it easier to understand (line 513, 556, 575, 620)

  1. The conclusion section should be written. The authors should describe and strengthen their findings and major results of the present work, not the significance of the present work.

Reply: conclusion was rewritten (lines 656-671)

Reviewer 3 Report

Comments and Suggestions for Authors

The aim of the manuscript was to investigate the efficacy of seed halopriming and acclimation in enhancing salt tolerance in bean plants. The study demonstrated that this approach significantly improves various aspects of bean growth, quality and yield under salt stress conditions. The main contributions of the paper lie in highlighting the practical utility of halopriming as a cost-effective technique to mitigate the negative impacts of salt stress on bean cultivation. Its strengths include providing valuable insights into the species-specific responses to priming, as well as comprehensive analyses of physiological and biochemical changes in response to salt stress and priming treatments.

The Abstract provides a concise overview of the study's objectives, methods, and key findings. It effectively highlights the importance of beans as a horticultural crop while emphasizing the challenges posed by salinity stress due to climate change. However, background is too broad. It could be improved by providing a more concise background, briefly giving a context and purpose of the study.

The Introduction provides a comprehensive overview of the importance of beans in human nutrition, emphasizing their nutritional value and health benefits. The authors effectively highlight the economic significance of beans and addresses the challenges posed by climate change and various stresses, including salinity, which adversely affect bean production.

The Materials and Methods section provides a detailed description of the experimental procedures employed to assess the impact of seed priming and acclimation on bean tolerance to saline conditions. The methodology is well-organized and encompasses various aspects, including plant material selection, growth conditions, physiological analyses, and statistical procedures. The authors provide clear details regarding the origin of the bean seeds and the methodology for evaluating seed halotolerance. However, specifying the specific concentrations of salt solutions used for the dose-response curve would improve methodological transparency. These data could be given as a supplementary material. Additionally, it would be beneficial to include information on the experimental design, such as the number of replicates and whether the experiments were repeated.

The results demonstrate the efficacy of seed priming in enhancing bean plant tolerance to salt stress through various physiological and biochemical mechanisms, ultimately leading to improved growth and productivity under saline conditions. However, few adjustments should be made to ensure consistency in presenting the results throughout the paper (Figures 5, 7, 9, 10; see comment below).

The discussion section provides an in-depth analysis of the results obtained from the study on enhancing salt tolerance in common bean plants through seed priming and acclimation. Overall, the discussion emphasizes the efficacy of seed priming as a promising strategy to enhance salt tolerance and improve productivity in common bean plants, addressing the challenges posed by climate change and salinity stress.

Specific comments:

Lines 125-126: Incomplete sentence.

Line 136: How is soil moisture determined and maintained (by measuring soil moisture, weighing)? What does humidity of 42% refer to (of soil water holding capacity, field capacity…)? This should be clarified for better understanding and replicability of the experiment.

Figures 5, 7, 9, 10: Review Figures and adjust the labels above the bars in the graph so that the letters used for labeling always start in the same way. For example, the letter "a" always represents the bar with the highest value.

Author Response

Seeds 2936464 Reply to Referee 3

The authors wish to thank the reviewer for the work and good suggestions to which we reply point by point

The aim of the manuscript was to investigate the efficacy of seed halopriming and acclimation in enhancing salt tolerance in bean plants. The study demonstrated that this approach significantly improves various aspects of bean growth, quality and yield under salt stress conditions. The main contributions of the paper lie in highlighting the practical utility of halopriming as a cost-effective technique to mitigate the negative impacts of salt stress on bean cultivation. Its strengths include providing valuable insights into the species-specific responses to priming, as well as comprehensive analyses of physiological and biochemical changes in response to salt stress and priming treatments.

The Abstract provides a concise overview of the study's objectives, methods, and key findings. It effectively highlights the importance of beans as a horticultural crop while emphasizing the challenges posed by salinity stress due to climate change. However, background is too broad. It could be improved by providing a more concise background, briefly giving a context and purpose of the study.

Reply: abstract was rewritten and simplified (line 17-30)

The Introduction provides a comprehensive overview of the importance of beans in human nutrition, emphasizing their nutritional value and health benefits. The authors effectively highlight the economic significance of beans and addresses the challenges posed by climate change and various stresses, including salinity, which adversely affect bean production.

The Materials and Methods section provides a detailed description of the experimental procedures employed to assess the impact of seed priming and acclimation on bean tolerance to saline conditions. The methodology is well-organized and encompasses various aspects, including plant material selection, growth conditions, physiological analyses, and statistical procedures. The authors provide clear details regarding the origin of the bean seeds and the methodology for evaluating seed halotolerance. However, specifying the specific concentrations of salt solutions used for the dose-response curve would improve methodological transparency. These data could be given as a supplementary material. Additionally, it would be beneficial to include information on the experimental design, such as the number of replicates and whether the experiments were repeated.

Reply: done (lines 108-109 and 123-126)

The results demonstrate the efficacy of seed priming in enhancing bean plant tolerance to salt stress through various physiological and biochemical mechanisms, ultimately leading to improved growth and productivity under saline conditions. However, few adjustments should be made to ensure consistency in presenting the results throughout the paper (Figures 5, 7, 9, 10; see comment below).

The discussion section provides an in-depth analysis of the results obtained from the study on enhancing salt tolerance in common bean plants through seed priming and acclimation. Overall, the discussion emphasizes the efficacy of seed priming as a promising strategy to enhance salt tolerance and improve productivity in common bean plants, addressing the challenges posed by climate change and salinity stress.

Specific comments:

Lines 125-126: Incomplete sentence.

Reply: done (line 120)

Line 136: How is soil moisture determined and maintained (by measuring soil moisture, weighing)? What does humidity of 42% refer to (of soil water holding capacity, field capacity…)? This should be clarified for better understanding and replicability of the experiment.

Reply: done (lines 135-137)

Figures 5, 7, 9, 10: Review Figures and adjust the labels above the bars in the graph so that the letters used for labeling always start in the same way. For example, the letter "a" always represents the bar with the highest value.

Reply: the same principle was used for all tables and graphs: the letter “a” was always associated with CTRL (primed or not primed)  

Reviewer 4 Report

Comments and Suggestions for Authors

1) Abstract needs to be rewritten. It is too long (340 words), while the Seeds journal recommends about 200 words maximum. In Abstract, such an extensive introduction is unnecessary. The Abstract should clearly outline the conclusions of the research and emphasize the contribution of the obtained results to the development of knowledge. See the instructions:  https://www.mdpi.com/journal/seeds/instructions#preparation

2) L 28: Please list the names of the tested bean cultivars.

3) Key words: instead of "acclimation" I suggest giving "acclimation to salt".

4) L 93: „Among four different varieties tested…” - please list these cultivars. Please explain why Black Turtle, Bola Roja and Cargamanto cultivars were selected for this research.

5) L 183, 195, 202: please enter the name of the spectrophotometer.

6) L 266, 478 and Table 7: seed weight per plant or per pod?

7) L 484-486; L 496-498: The sentence "Significant differences within the same group (...) are reported with different letters in the column" should be corrected. From the analysis of the data in Table 7 and 8, it appears that statistical analysis was conducted separately for CTRL and 40 mM NaCl.

8) Feature values in Table 7 should be reported with greater accuracy, as Table 8 has done. In Table 7, the values of the feature "n. pods/plant" in the Priming solution 40 mM NaCl and 0 and 40 mM NaCl variants are "2", and they were assigned to homogeneous groups a and ab, respectively, due to excessive rounding of the values.

9) L 501-506: This part of the text should be included in the Introduction.

Author Response

Reply to Referee 4

The authors wish to thank the reviewer for the work and good suggestions to which we reply point by point

Comments and Suggestions for Authors

  • Abstract needs to be rewritten. It is too long (340 words), while the Seeds journal recommends about 200 words maximum. In Abstract, such an extensive introduction is unnecessary. The Abstract should clearly outline the conclusions of the research and emphasize the contribution of the obtained results to the development of knowledge. See the instructions:  https://www.mdpi.com/journal/seeds/instructions#preparation

Reply : abstract was rewritten (line 17-30)

  • L 28: Please list the names of the tested bean cultivars.

Reply: done (line 22)

  • Key words: instead of "acclimation" I suggest giving "acclimation to salt".

Reply: done (line 31)

  • L 93: “Among four different varieties tested…” - please list these cultivars. Please explain why Black Turtle, Bola Roja and Cargamanto cultivars were selected for this research.

Reply: done (line 84-87)

  • L 183, 195, 202: please enter the name of the spectrophotometer.

Reply: done (lines 185, 195 and 202)

  • L 266, 478 and Table 7: seed weight per plant or per pod?

Reply: done (line 266, and Table 7)

  • L 484-486; L 496-498: The sentence "Significant differences within the same group (...) are reported with different letters in the column" should be corrected. From the analysis of the data in Table 7 and 8, it appears that statistical analysis was conducted separately for CTRL and 40 mM NaCl.

Reply: we apologize but we cannot understand this comment

  • Feature values in Table 7 should be reported with greater accuracy, as Table 8 has done. In Table 7, the values of the feature "n. pods/plant" in the Priming solution 40 mM NaCl and 0 and 40 mM NaCl variants are "2", and they were assigned to homogeneous groups a and ab, respectively, due to excessive rounding of the values.

Reply: done (line 473, Table 7)

  • L 501-506: This part of the text should be included in the Introduction.

Reply: most of the information is already present in the introduction (line 34-54)